# Transport of Heavy Metals Pb(II), Zn(II), and Cd(II) Ions across CTA Polymer Membranes Containing Alkyl-Triazole as Ions Carrier

**DOI:** 10.3390/membranes12111068

**Published:** 2022-10-29

**Authors:** Elżbieta Radzymińska-Lenarcik, Sylwia Kwiatkowska-Marks, Artur Kościuszko

**Affiliations:** 1Institute of Mathematics and Physics, Bydgoszcz University of Science and Technology, Al. Prof. S. Kaliskiego St. 7, PL-857-96 Bydgoszcz, Poland; 2Faculty of Chemical Technology and Engineering, Bydgoszcz University of Science and Technology, Seminaryjna St. 3, PL-853-26 Bydgoszcz, Poland; 3Faculty of Mechanical Engineering, Bydgoszcz University of Technology, Al. Prof. S. Kaliskiego 7, PL-857-96 Bydgoszcz, Poland

**Keywords:** polymer inclusion membrane (PIM), lead, zinc, cadmium, separation, heavy metals

## Abstract

The polymer membranes of cellulose triacetate -*o*-NPPE-1-alkyl-triazole (alkyl= hexyl, octyl, decyl) were characterized by non-contact atomic force microscopy (AFM). The influence of membrane morphology on transport process was discussed. 1-Alkyl-triazole derivatives are new cheap compounds that have the ability to bind metal ions in an acidic medium. These membranes were used for the investigation of the facilitated transport of Zn(II), Cd(II), and Pb(II) ions from an aqueous nitrate feed phase. The initial flux values of metal ions transport depend on the type of carrier used. The maximum value of the initial flux for Zn(II) ions was equal to 12.34 × 10^−6^ molm^−2^s^−1^ (for PIMs with 1-decyltriazole). In the case of Zn(II) and Cd(II) ions as the hydrophobicity of the carrier increases, the separation coefficients S_Zn(II)/Cd(II)_ slightly increase from 1.8 to 2.4, while for Zn(II) and Pb(II) ions separation coefficients S_Zn(II)/Pb(II)_ decrease. The highest recovery factors (RF) were found for Zn(II) ions (c.a. 90%). The RF values of Cd(II) ions increase from 58% to 67%. The highest RF value for Pb(II) is 30%. The rate-limiting step in the transport of Zn(II), Cd(II) and Pb(II) ions across PIMs with 1-alkyltriazole may be the diffusion coefficient of the carrier-cation complex. The AFM images show that the distribution of the carrier in the tested membranes is homogeneous over the entire surface. The roughness values determined for PIMs with alkyltriazole are slightly higher than the roughness of PIM with the commercial carrier, for example D2EHPA.

## 1. Introduction

The demand for metals has been increasing for many years. Heavy metals are widely used in metallurgy, electroplating, chemical, tannery, petrochemicals and paper manufacturing industries, etc.: The use of heavy metals for industrial production is the main reason why these metals are present in wastewater, sewage and waste [1]. Without proper waste water treatment, heavy metals will be released into water bodies and cause severe damage to the environment and ecosystem [2,3,4,5].

Among heavy metals, cadmium, lead, and zinc are considered the most toxic and hazardous to the environment [6,7,8]. They are currently implemented in a significant number of industries such as the production of cables, batteries, pigments, paints, steels and alloys, as well as the metal, glass, and plastic industries. The discharge of these industries causes the contamination of the aquatic environment by these heavy metals [9]. Metals removed from the sewage can be successfully reused in industry, therefore sewage and metal-bearing waste have become a secondary source of metals. Conventional methods of wastewater treatment, such as precipitation [10], ion exchange [11], flotation [12] or coagulation, require the use of large volumes of reagents, resulting in the formation of large amounts of waste containing heavy metals. For example, industrial wastewaters containing 0.1 g/dm^3^ of copper, cadmium, or mercury ions give 10-, 9-, and 5-times larger amounts of sediments, respectively [13]. Also, liquid-liquid extraction is widely used in the separation of heavy metals. However, due to the toxicity of the water-insoluble solvents used, these methods are very risky, especially when used on an industrial scale [14]. Therefore, it is necessary to search for alternative, cheap and ecological methods of purifying sewage and wastewater from various types of pollutants, especially from heavy metal ions.

Over the decades, it has been proven that liquid membranes, in particular polymer inclusion membranes (PIM), are a better alternative than solvent extraction methods for the separation and recovery of various metal ions [14,15,16,17,18]. The advantage of using PIMs is the possibility of simultaneous extraction and re-extraction and their greater durability and stability compared to liquid membranes [19]. The PIMs are formed by evaporating a solution containing a polymer matrix, a plasticizer, and a carrier [20]. In polymer inclusion membranes, the carrier is not washed out of the matrix, and in addition, these membranes do not contain any solvent. These membranes are characterized by a much longer life, better mechanical properties and chemical resistance compared to liquid membranes (LM) [21,22]. These types of membranes are increasingly used to remove a number of heavy metals from wastewater, including Cu, Zn, Pb, Ni, Co, Cd [1,2,14,23].

New carriers, that selectively bind heavy metal ions in order to effectively remove them from wastewater, are constantly sought after. Triazoles are nitrogen-containing heterocycles. 1,2,4-Triazoles and their amino derivatives have also been studied as extractants of Co(II), Ni(II), Zn(II), Cd(II) ions [24]. Also, the effectiveness in removing Cu(II) from the Cu(II)-Ni(II) mixture during solvent extraction with 1-alkyl-1,2,4-triazoles has been investigated [25,26]. 

The present article deals with a competitive transport of zinc(II), lead(II), and cadmium (II) ions from a dilute aqueous solutions using PIM with 1-alkyltriazoles. The values of three parameters: initial flux (J_0_), selectivity coefficient (S_M(1)_/_M(2)_), and recovery factor of a given metal after 24 h (RF) were selected for the comparative analysis of the transport process. The aim of the study was also to investigate the influence of the physical properties of PIMs on the efficiency of separation of the studied ions.

Cadmium, lead, and zinc are considered the most toxic and hazardous to the environment [23,24,25]. They are currently implemented in a significant number of industries such as cables, batteries, pigments, paints, steels and alloys, metal, glass, and plastic industries. The discharge of these industries causes the contamination of the aquatic environment by these heavy metals [26].

## 2. Experimental

### 2.1. Reagents

Analytical grade chemical reagents: zinc(II), cadmium(II), and lead(II) nitrates, sulphuric acid, and tetramethylammonium hydroxide were purchased from (POCh, Gliwice, Poland). Solutions of investigated metal ions were prepared by dissolving appropriate amounts of Zn(NO_3_)_2_·6H_2_O, Cd(NO_3_)_2_·4H_2_O, and Pb(NO_3_)_2_ in deionized water.

Analytical grade organic reagents, i.e., cellulose triacetate (CTA), o-nitrophenyl pentyl ether (o-NPPE) and dichloromethane (all from Sigma-Aldrich Company, Poznan, Poland) were used without further purification.

1-Alkyl-1,2,4-triazole derivatives (Table 1) were synthesized by prof. A. Skrzypczak (Poznan University of Technology, Poznan, Poland) by the alkylation reaction of 1,2,4-triazole according to the following Equation (1).

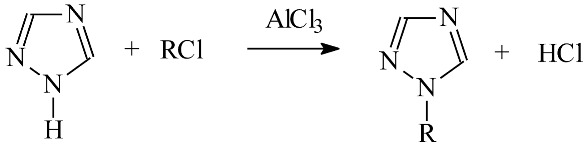
(1)

Physical properties of the 1-alkyltriazoles are listed in the Table 1.

### 2.2. Equipment 

Metal ions concentrations in aqueous phases were analyzed with AAS 240FS Spectrometer, Agilent, Santa Clara, CA, USA (AAS—atomic absorption spectroscopy). Measurements were made for the following emission lines of the analyzed elements: 213.9 nm, 228.8 nm, and 247.6 nm for Zn, Cd, and Pb respectively. 

The pH-meter (PHM 250 (Radiometer, Copenhagen, Denmark) equipped with a glass-calomel combination electrode C 2401-8 (Radiometer, Copenhagen, Denmark) was calibrated using commercial buffer solutions (Radiometer, Copenhagen, Denmark) having a pH of 4.01 ± 0.01, 7.00 ± 0.01 and 9.21 ± 0.01.

The thickness of the PIM was measured using a digital micrometer (Panametrics^®^ Magna-Mike^®^ 8500 (San Diego, CA, USA)) with an accuracy of 0.1 µm. 

A surface characterization study of the PIMs was performed by atomic force microscopy (AFM) using Atomic-force MultiMode Scanning Probe Microscope IIIa (Digital Instruments Vecco Metrology Group, Santa Barbara, CA, USA).

The Netzsch TG 209 F1 Libra thermogravimetric analyzer (Selb, Germany) was used in thermogravimetric studies.

Membrane spectrophotometric studies were performed using a Bruker Invenio R Infrared Spectrophotometer (Ettlingen, Germany) equipped with a broadband BeamSplitter and the ATR Quest attachment by Specac (Orpington, UK). The research was carried out in the wavenumber range from 3750 to 300 cm^–1^.

### 2.3. Polymer Inclusion Membrane Preparation

PIMs were prepared as reported in the earlier paper [27,28,29,30,31]. The feed phase was a 3-component aqueous solution of Pb(II), Zn(II), and Cd(II) ions with a concentration of C_0_,_M_ =0.001 M of each ion. The feed phase pH was kept constant (pH = 4.0) by adding tetramethylammonium hydroxide and controlled by pH meter. The receiving phase was 0.01 M HCl. At the receiving phase, metal ions concentrations were measured. The membrane film (at surface area of 4.9 × 10^−4^ m^2^) was tightly clamped between two cell compartments. Both, i.e., the source and receiving aqueous phases (45 cm^3^ each) were mechanically stirred at 600 rpm. Metal concentration was determined by taking small samples (0.1 cm^3^ each) of the aqueous receiving phase at different times. 

Membranes with a composition of 45% o-NPPE, 30% CTA and 25% 1-alkyltriazole **1**–**3** were used in the research, because, according to our previous research [23,27,28,30,31,32], such a membrane composition guaranteed the most effective transport of metal ions.

### 2.4. Parameters Characterizing the Transport Process

The process of transport across PIMs is characterized by the initial flux (*J_o_*), selectivity coefficient (S), and recovery factor (RF) [33,34].

The kinetics of metal ions transport across membranes was described by equation:(2)$$ln(\frac{c_{t}}{c_{0}})=-kt$$
where *c_t_* and *c_o_* are the metal ions concentration (M) in the feed phase at a given time, and the initial metal ions concentration, respectively; *k* is the rate constants (s^−1^), t is the time of transport (s).

The permeability coefficient (*P*) was calculated from:(3)$$P=-\frac{V}{A}k$$
where *V* is the volume of the aqueous feed phase, and *A* is an effective area of the membrane.

The initial flux (*J*_0_) is defined as:(4)$$J_{0}=Pc_{0}$$

The selectivity coefficient (*S*) was defined as the ratio of initial fluxes for *M*1 and *M*2 metal ions, respectively:(5)$$S=\frac{J_{0,M1}}{J_{0,M2}}$$

To describe the efficiency of metal removal from the feed phase, the recovery factor (*RF*) was calculated:
(6)$$RF=\frac{c_{0}-c_{t}}{c_{0}}\cdot 100\%$$

## 3. Results and Discussion

### 3.1. Membrane Characteristics 

The thickness of membranes before and after transport was found to be the same and were 30, 31, 33 μm for **1**, **2**, **3**, respectively.

The analysis of surface pore characteristics of the polymer membrane was made using the NanoScope v.5.12 AFM image processing program, which enabled the calculation of PIM’s roughness (*R_q_*).

Figure 1 shows the AFM images of PIMs with 1-alkyltriazole (**1**–**3**) as the carrier in three-dimensional form with format of 5.0 × 5.0 µm. The distribution of the carrier in the investigated membrane after evaporation of the dichloromethane is homogeneous on the entire surface.

The roughness (*R_q_*) parameter is the standard deviation of the z values within the box cursor and is calculated as:(7)$$R_{q}=\sqrt{\frac{\sum^{\;}{\left(z_{i}\right)}^{2}}{n}}$$

The average roughness (*R_q_*) of the membrane was calculated using atomic force microscopy (AFM) for *n* different sites, and they are shown in Table 2.

As seen in Table 2 the roughness of the membranes increases with increasing length of the alkyl substituent in the carrier molecule higher than the commercial carrier, D2EHPA (4.7 nm) that was used by Salazar-Alvarez [35]. However, the roughness values for CTA-o-NPPE-alkyltriazole membranes are comparable to the roughness found for PIM with alkylimidazoles (3.7–7.2 nm) [23] and with an imidazole derivative of azothiacrown ethers (3.3–5.3 nm) [36].

#### 3.1.1. Thermal Stability of PIM with 1-Alkyltriazole

The CTA-*o*-NPPE membrane with 1-hexyltriazole was also tested for their thermal stability. About 20 mg of membrane were heated at 10 °C/min under nitrogen from 25 °C to 800 °C. Figure 2 shows thermograms of CTA-o-NPPE membranes with and without 1-hexyltriazole (**1**) before and after ion transport.

As seen from Figure 2 the degradation of CTA-o-NPPE membrane with 1–hexyltriazole proceeds in two steps. The ranges of decomposition temperatures are summarized in Table 3.

PIM made of CTA-o-NPPE with 1–hexyltriazole shows high thermal stability (up to approx. 170 °C) (Figure 2). For these membranes, the first step of degradation occurs at 249.8–277.0 °C, while the second one at 343.4–367.8 °C. The corresponding weight losses are within ranges of 28.20–80.61% and 16.37–44.33%, respectively.

As reported in the literature, the degradation of a CTA membrane occurs in two steps; the first one over a range of 292–320 °C (main step) and the other one over a range of 450–476 °C (the charring of products) [37,38,39].

#### 3.1.2. FT-IR Analysis of the PIM with 1-Hexyltriazole

The membrane with 1-hexyltriazole was analyzed using FT-IR spectroscopy. The spectra are presented in Figure 3.

The interpretation of IR spectra was made using IRPal 2.0 program. Table 4 shows indicated bonds of signals recorded in Figure 3.

### 3.2. Separation of Zn(II) from Zn(II)-Cd(II)-Pb(II) Mixture 

In the first series of experiments, competitive transport of Zn(II), Cd(II), and Pb(II) ions from aqueous nitrate solutions containing the metal species at concentration 0.001 M across CTA polymer membrane with alkyltriazole as the ionic carrier and o-NPPE as the plasticizer into 0.01 M HCl was investigated.

In the absence of an ion carrier in the PIM, i.e., when the membrane contained only CTA and o-NPPE, no significant flow of metal ions was detected.

The relationship between ln(*C_t_/C_o_*) and time for the transport of Zn(II), Cd(II), and Pb(II) ions across PIMs containing **1**–**3** triazole derivatives is shown in Figure 4. 

The relationship ln(*C_t_/C_o_*) = f(t) was linear (Figure 4), which was confirmed by high values of determination coefficients (R^2^) (above 0.98). Standard deviation of rate constant was determined below 5%.

As shown in Figure 4, transport can be described by first-order kinetics in relation to the concentration of metal ions. It is in agreement with the mathematical model proposed by Danesi [40]. 

The initial fluxes, selectivity order and selectivity coefficients for competitive transport of Pb(II), Zn(II), and Cd(II) ions across PIMs with 1-alkyl-1,2,4-triazole (**1**–**3**) are calculated from Equations (3) and (4), and are presented in Table 5.

As can be seen from Table 5, for all carriers, the initial zinc ion flux has the highest value, which means that Zn (II) ions were transported at the highest rate. Lead(II) ions were transported with the lowest. 

The initial flux values of metal ions transport also depend on the type of carrier used and increases in the order **1** < **2** < **3**. The observed trend increases with an increase in the hydrophobicity of the carrier (with an increase in the length of the -R groups in the carrier molecule). The maximum value of the initial flux for Zn(II) ions was equal to 12.34 × 10^−6^ molm^−2^s^−1^ (for **3**). 

Comparing the values of the initial fluxes (Table 5) with the roughness values (Table 2) of the membranes, it can be concluded that the increase in roughness increases the transport of each metal ion.

In the case of zinc and cadmium ions as the hydrophobicity of the carrier increases, the values of separation coefficients S_Zn(II)/Cd(II)_ slightly increase from 1.8 for carrier **1** to 2.4 for **3** while for zinc and lead ions the values of separation coefficients S_Zn(II)/Pb(II)_ decrease. The highest value of the separation coefficients S_Zn(II)/Pb(II)_ equal to 16.2 was obtained for carrier **1**.

The Figure 5 presents the proposed mechanism of the metal ions transport across PIMs with alkyltriazole.

At the interface of the feed phase and the PIM surface, metal ions bind to the carrier molecules (alkyl triazoles) present in the membrane to form complex compounds. The complexes diffuse to the opposite surface of the membrane where the metal ions are released into the receiving phase.

The Pb(II), Zn(II), and Cd(II) ions with alkyltriazole (L) form 6-coordinate octahedral complexes [ML_6_]^2+^. In the case of Zn(II) and Cd(II) ions, an additional phenomenon is the ease of changing the coordination number from 6 to 4, and thus the geometry of the coordination sphere from octahedron to tetrahedron [29,41,42]. This is illustrated by the Equation (8).

For Zn(II) and Cd(II):  [M(H_2_O)_6_]^2+^ + nL ↔ [M(H_2_O)_4−n_L_n_]^2+^ + (n + 2) H_2_O(8)

The transport mechanism presented in Figure 5 can be described as follows:Transport:

at the interface feed phase/ membrane: [M]^2+^_(aq)_ + nL ↔ [ML_n_]^2+^_(membrane)_
n = 4 for Cd(II) and Zn(II), n = 6 for Pb(II)

at the interface membrane/ receiving phase: [ML_n_]^2+^_(membrane)_ ↔ [M]^2+^_(aq/HCl)_ + nL_(membrane)_

2.Back-transport:

at the interface receiving phase/membrane: L_(membrane)_ + H^+^_(aq/HCl)_ ↔ HL_(membrane)_

at the interface membrane/feed phase: HL_(membrane)_ ↔ L_(membrane)_ + H^+^_(aq)_

The differences in the transport speed of the examined ions across the PIM can be explained by comparing the influence of their diameter on the values of the initial fluxes (Figure 6).

The diameters of the Zn(II), Cd(II), and Pb(II) are 1.48Å, 1.90Å, and 2.36Å, respectively [43]. As the cation diameter increases, the value of the initial ion fluxes decreases (Figure 6).

The transport of metal ions is influenced by both the size of the cation and the size of the carrier molecule.

### 3.3. Membrane Diffusion Coefficients of Zn(II), Cd(II), and Pb(II) Ions across PIMs with 1- Alkyltriazole

Figure 7 shows the graphs of the correlation (C_o_–C_t_) as a function of time for the transport of Zn(II), Cd(II), and Pb(II) ions across PIM with 1-alkyltriazole **1**–**3**.

According to Tor et al. [44], the diffusion coefficients of metal ions (D_o_) can be calculated from the equation:
D_o_ = d_o_/Δ_o_
(9)
where: d_o_—the thickness of the membrane and Δo could be evaluated by plotting (C_o_–C_t_) vs. time. 

Obtained values of diffusion coefficients are presented in Table 6.

Values of diffusion coefficient determined in this study are comparable with these presented in literature data for different membranes. They are in the range from 10^−12^ to 10^−6^ cm^2^/s and show that the limiting step of the process is the transfer of metal complex across membrane barrier [35,44]. The value of the diffusion coefficient of M(II)-carrier species of 3.12 × 10^−11^–2.38 × 10^−7^ cm^2^/s is comparable to PIMs with 1-alkylimidazole for which the values of diffusion coefficients range from 10^−12^ to 10^−8^ cm^2^/s [23].

### 3.4. Recovery of Metal

In order to describe the efficiency of metal removal from the feed phase, the recovery factor (RF) was calculated from Equation (5). Figure 8 shows the values of the recovery factor Zn(II), Cd(II), and Pb(II) ions from the feed phase during the 24-hrs transport across PIMs with 1-alkyl-triazoles.

As shown in Figure 8, the values of recovery factors (RF) depend on the carrier used in the membrane. The RF values for Zn(II) and Cd(II) ions increase with increasing hydrophobicity of the carrier molecule, hence they are the highest for 1-decyltriazole (**3**) in contrast to the RF value for Pb(II) ions, which is the lowest for this carrier. The highest recovery factors (RF) were found for Zn(II) ions (c.a. 90%). The RF values of Cd(II) ions are 58%, 63%, and 67% for **1**, **2**, and **3**, respectively.

The lowest RF values were obtained for Pb(II) ions, which are the slowest transported by this type of membrane. The highest RF value for Pb(II) is 28% for carrier **1**. 

## 4. Conclusions

1-Alkyl triazole derivatives are new cheap compounds that have the ability to bind metal ions in an acidic medium. They can be used as carriers in PIMs.

The 1-alkyltriazoles under study can be used for separation of the Zn(II) ions from an equimolar mixture of the Zn(II), Cd(II) and Pb(II) ions during transport across PIMs. The transport of Zn(II), Cd(II) and Pb(II) ions can be described by first-order kinetics in relation to the concentration of metal ions.

The initial flux values of metal ions transport depend on the type of carrier used and increases with the increase in the hydrophobicity of the carrier molecule in the order **1** < **2** < **3**. The maximum value of the initial flux for Zn(II) ions was equal to 12.34 × 10^−6^ molm^−2^s^−1^ (for **3**). 

In the case of zinc and cadmium ions as the hydrophobicity of the carrier increases, the values of separation coefficients S_Zn(II)/Cd(II)_ slightly increase from 1.8 for carrier **1** to 2.4 for **3** while for zinc and lead ions the values of separation coefficients S_Zn(II)/Pb(II)_ decrease. 

The highest value of the separation coefficients S_Zn(II)/Pb(II)_ equal to 16.2 was obtained for carrier **1**. The highest recovery factors (RF) were found for Zn(II) ions (c.a. 90%). The RF values of Cd(II) ions are 58%, 63%, and 67% for **1**, **2**, and **3**, respectively. The highest RF value for Pb(II) is 30% for carrier **2**.

The rate-limiting step in the transport of Zn(II), Cd(II) and Pb(II) ions across PIMs with 1-alkyltriazole may be the diffusion coefficient of the carrier-cation complex.

The values of the initial fluxes of all metal ions transported are low, which means that membranes of this type can be used in industry in the next step of wastewater treatment when the concentrations of toxic metals (Zn, Cd and Pb) are low.

## Figures and Tables

**Figure 1 membranes-12-01068-f001:**
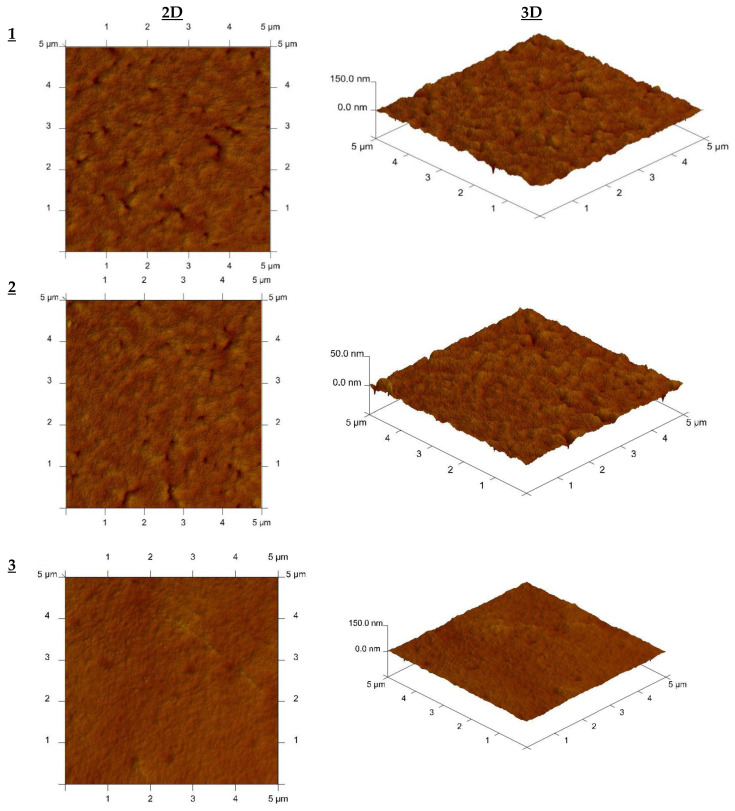
AFM pictures of the PIMs with 1-alkyltriazole; **1**—1-hexyltriazole, **2**—1-octyltriazole, **3**—1-decyltriazole.

**Figure 2 membranes-12-01068-f002:**
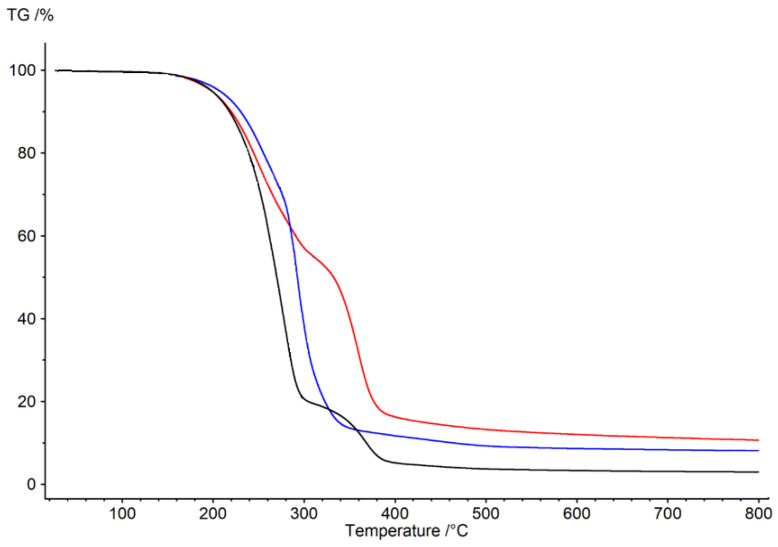
TG curves for membranes CTA-o-NPPE (black line); CTA-o-NPPE with 1–hexyltriazole before process (blue line) and CTA-o-NPPE with 1–hexyltriazole after process (red line).

**Figure 3 membranes-12-01068-f003:**
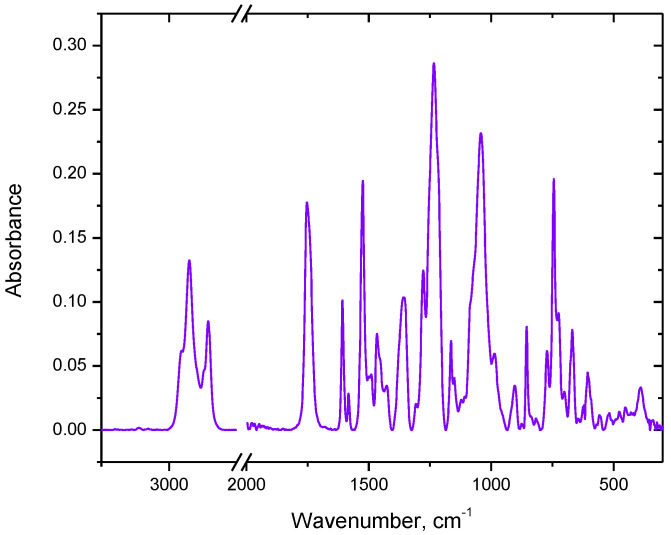
FT-IR spectra of PIM with 1–hexyltriazole.

**Figure 4 membranes-12-01068-f004:**
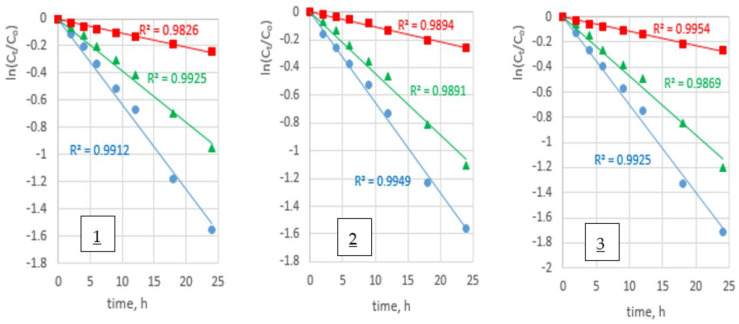
Kinetics curves for metal ions transport of (●) Zn(II), (▲) Cd(II), and (■) Pb(II) ions from equimolar mixture of metal ions across PIMs with 1–hexyltriazole (**1**), 1–octyltriazole (**2**), and 1–decyltriazole (**3**); feed phase: CM = 0.001 M each metal ion, pH = 4.0; receiving phase: 0.01 M HCl.

**Figure 5 membranes-12-01068-f005:**
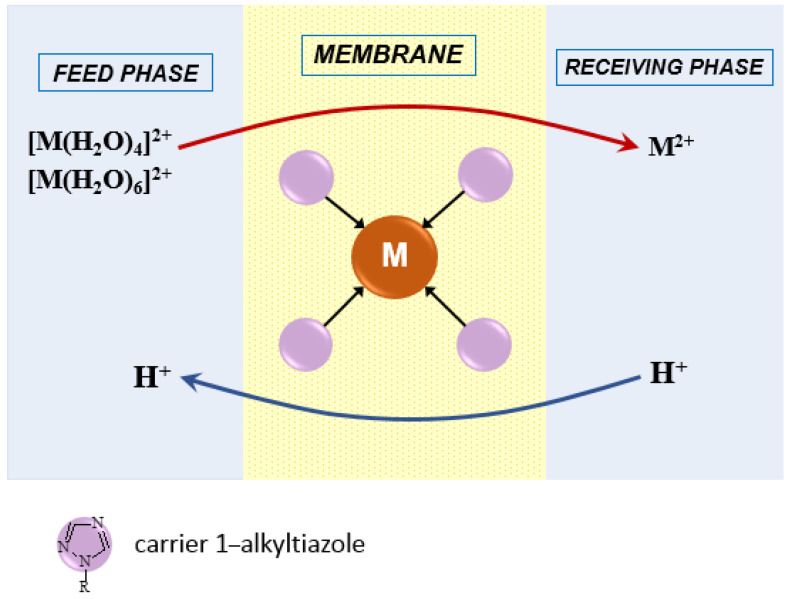
Schematic transport of metal ions across PIMs with alkyltriazole.

**Figure 6 membranes-12-01068-f006:**
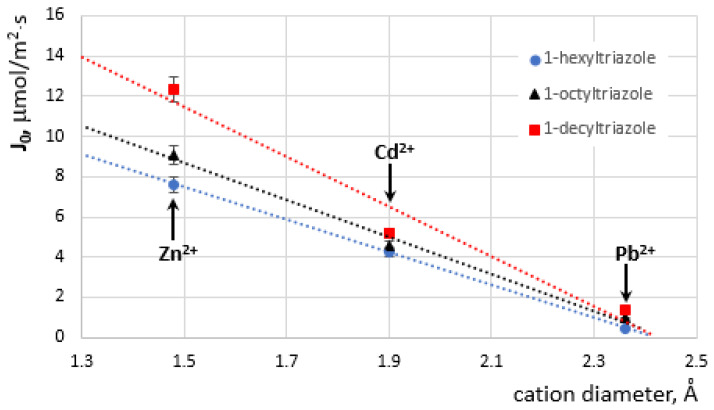
Dependence of the initial flux value of Zn(II), Cd(II), and Pb(II) ion transport by PIM containing 1–hexyltriazole (**1**), 1–octyltriazole (**2**), and 1–decyltriazole (**3**) as carriers.

**Figure 7 membranes-12-01068-f007:**
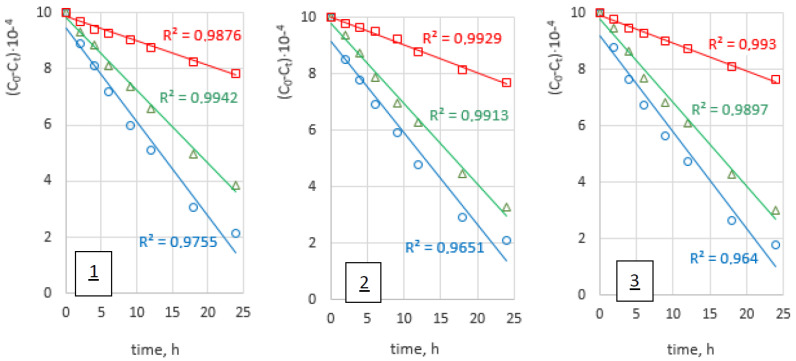
Relation of (Co–Ct) plotted vs. time for (○) Zn(II), (△) Cd(II), and (□) Pb(II) ions transport across PIMs with 1–hexyltriazole (**1**), 1–octyltriazole (**2**), and 1–decyltriazole (**3**).

**Figure 8 membranes-12-01068-f008:**
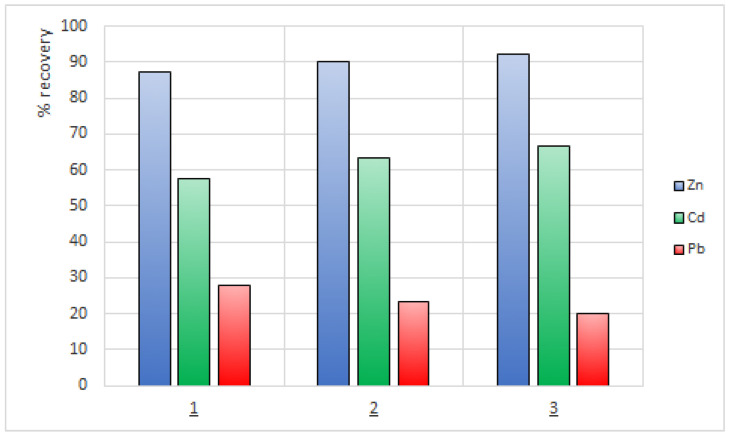
Recovery factor (RF) of zinc, cadmium and lead during transport across PIMs with 1-hexyltriazole (**1**), 1-octyltriazole (**2**), and 1-decyltriazole (**3**).

**Table 1 membranes-12-01068-t001:** Properties of 1-alkyl-triazoles.

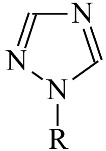	**No.**	**R =**	**Compound**	**Boiling Point, °C,** **at Pressure 1 hPa**
	**1**	–C_6_H_13_	1-hexyl-1,2,4-triazole	179–181
	**2**	–C_8_H_17_	1-octyl-1,2,4-triazole	185–188
	**3**	–C_10_H_21_	1-decyl-1,2,4-triazole	216–218

**Table 2 membranes-12-01068-t002:** The average roughness for PIM with 1-alkyltriazole; **1**–1-hexyltriazole, **2**–1-octyltriazole, **3**–1-decyltriazole.

Polymer Inclusion Membranes with 1-Alkyl-Triazole			
**Carrier**	**1**	**2**	**3**
**Roughness**, nm	4.52	4.89	5.21

**Table 3 membranes-12-01068-t003:** Degradation temperatures and weight losses of CTA-o-NPPE membrane with 1-hexyltriazole.

The Composition of Membrane	The First Step		The Second Step	
	Temp. °C	Weight Loss, %	Temp. °C	Weight Loss, %
CTA-o-NPPE	277.0	80.61	367.8	16.37
CTA-o-NPPE– 1-hexyltriazole before process	268.0	28.20	343.4	44.33
CTA-o-NPPE– 1-hexyltriazole after process	249.8	44.91	359.6	44.33

**Table 4 membranes-12-01068-t004:** Indicated bonds in PIM with 1-hexyltriazole.

Range of Wavenumbers, cm^−1^	Indicated Bonds
2925–2750	C-H, N-H, O-H
1755–1450	C-C, C-O, C-N
1660–1490; 1390–1260	N-O
1635–1470	C=C, C=N (triazole ring)
870–840	C-N
770–665	C-H
480–390	-NO_2_

**Table 5 membranes-12-01068-t005:** Initial fluxes, selectivity order and selectivity coefficients for competitive transport of Pb(II), Zn(II), and Cd(II) ions across PIMs with 1-alkyl-1,2,4-triazole (**1**–**3**), membrane: 45% o-NPPE, 30% CTA, and 25% of 1-alkyltriazole **1**–**3**.; feed phase: 0.001 M each metal ion, pH = 4.0; receiving phase: 0.01 M HCl.

Carrier	Metal Ions	*J*_0_, μmolm^−2^·s^−1^	Selectivity Orders and Selectivity Coefficients S_Zn(II)/M(II)_ S_Pd(II)/M(II)_
**1**	Zn(II)	7.62	Zn(II) > Cd(II) > Pb(II) 1.8 16.2
	Cd(II)	4.25	
	Pb(II)	0.47	
**2**	Zn(II)	9.05	Zn(II) > Cd(II) > Pb(II) 2.0 9.5
	Cd(II)	4.58	
	Pb(II)	0.95	
**3**	Zn(II)	12.34	Zn(II) > Cd(II) > Pb(II) 2.4 8.8
	Cd(II)	5.19	
	Pb(II)	1.41	

**Table 6 membranes-12-01068-t006:** Diffusion coefficients for competitive transport of Zn(II), Cd(II), and Pb(II) ions through PIMs with 1-alkyltriazole (**1**–**3**).

Carrier	Metal Ion	Δ_o_, s/m	D_o_, cm^2^/s
**1**	Zn(II)	103.04	2.38 × 10^−7^
	Cd(II)	176.08	3.52 × 10^−8^
	Pb(II)	1256.14	2.73 × 10^−11^
**2**	Zn(II)	128.25	2.79 × 10^−7^
	Cd(II)	196.13	3.86 × 10^−8^
	Pb(II)	1287.06	2.91 × 10^−11^
**3**	Zn(II)	135.42	2.94 × 10^−7^
	Cd(II)	216.17	3.95 × 10^−8^
	Pb(II)	1298.24	3.12 × 10^−11^

## Data Availability

Not applicable.

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
