# Peer review of "Transport of Heavy Metals Pb(II), Zn(II), and Cd(II) Ions across CTA Polymer Membranes Containing Alkyl-Triazole as Ions Carrier"

_membranes, 2022, doi:10.3390/membranes12111068_

Round 1
Reviewer 1 Report
attached file

Author Response
Thank you for your time and valuable substantive comments.
- The importance of new membrane materials was highlighted. The paragraphs were connected. The following has been added to the introduction:
In polymer inclusion membranes, the carrier is not washed out of the matrix, and in addition, these membranes do not contain any solvent. These membranes are characterized by a much longer life, better mechanical properties and chemical resistance compared to liquid membranes (LM).
- 1-Alkyl triazole derivatives have been used as carriers in PIMs for the separation of Pd-Ni-Zn ion mixtures. Information is provided in line 65-67.
- The quality of the figures was improved.
- We have not combined this data because we find this presentation to be easier to read.
- Table 2 shows the mean values of the roughness measurements at the N points of the membrane. The measuring points cover the entire surface of the diaphragm. The roughness was calculated using the NanoScope v.5.12 AFM image processing program. The text has been supplemented.
- We agree with the Reviewer. The speed of transport is influenced by many factors, including the volume of the molecule.
Reviewer 2 Report
1. The Introduction section should further highlight the novelty of their study, such as the new membrane materials. Besides, there are too many paragraphs in the Introduction. Some of them are just one sentence. The authors should consider merging them.
2. Were the 1-Alkyl-triazole derivatives previeously reported in other papers? If so, the authors should include citations.
3. The quality of the figures should be improved. For example, the texts in the Figure 4, 6, and 7 are not clear.
4. The information shown in Table 2 could be embedded in Figure 1. Similarly, the information shown in Table 4 could be embedded in Figure 3.
5. The authors concluded that the increase in roughness increases the transport of each metal ion (Lines 268-270). However, the roughness difference among the three membranes (Table 2) may not be statistical significant. Their values were close to each other. The authors should conduct statistical analysis to compare the roughness.
6. What is the role of the -R groups in the carriers? The authors suggested that the hydrophobicity of the -R group affected the ion flux (Lines 264-267). However, it is also possible that a longer -R group occupies a larger volume, which increases the membrane free volume and enhences mass transfer.
Author Response
Thank you for your time and valuable substantive comments.
In the tested systems, the hydrophobicity of the carrier molecule and the diameter of the transferred ion affect the transport speed. Both the increase in hydrophobicity of the alkyltriazole molecule and the decrease in the diameter of the metal cation resulted in an increase in the size of the initial flux.
Title. Title was corrected.
Abstract. Corrected as suggested by the Reviewer.
- Introduction. Corrected as suggested by the Reviewer.
The sentences concerning the biological activity of triazole have been deleted.
Cu was separated from the binary mixture of Cu-Ni ions. The sentence was corrected.
- Experimental. Corrected as suggested by the Reviewer.
Metal concentration was determined by taking small samples of the aqueous receiving phase at different times.
- Results and discussion. Corrected as suggested by the Reviewer.
3.1.1. No crystals or inclusions of solvent particles were seen in the entire area that was visible under the AFM microscope, therefore the membrane surface was found to be homogeneous on the entire surface.
3.1.2. The TG studies were performed only for the 1-hexyltriazole membrane, because this membrane was the most effective in the separation of Zn from the Zn-Cd-Pb mixture.
Previous studies on the thermal stability of membranes show that the decomposition of each CTA-based membrane takes place in two steps. We did not expect a different result, but there may be exceptions.
3.1.3. The FTIR of all membranes is very similar, therefore the FTIR analysis for the most effective membrane is provided.
3.2. Separation of Zn(II) from Zn-Cd-Pb mixture
Corrected as suggested by the Reviewer. The text from lines 223-230 was moved to the introduction section.
3.5. J0 characterizes the transport of metal ions in the initial step of the process and does not always correspond to the final result. RF illustrates the transport efficiency and gives the amount of metal ions recovered after a 24-hour process.
Reviewer 3 Report
Overall:
- Need major English correction
- Can further improve by adding discussions to link membrane thermal stability and FTIR results with transport of metal ions across PIM.
- Results indicated that the use of more hydrophobic triazole carrier produced rougher membrane surface, which helped increase the transport flux. It was also mentioned that transport flux was affected by the size of metal cations. How about the role of chemical interaction between metal ions and the different triazole carriers? It will be helpful to include more discussion from chemical perspective such as information/figure/equation that illustrate the chemical bonding between the carrier and metal ion; how does different alkyl chain length triazole carriers chemically affect metal ions transport and separation? (based on current manuscript, it seems that selectivity in transport was due to metal cation size rather than the use of triazole carriers)
Title needs correction:
Transport of heavy metals Pb(II), Zn(II), and Cd(II) ions across CTA polymer membranes containing alkyl-triazole as ions carrier
Please check unit of flux; should be mol/m2.s or mol/m2-s
Abstract:
Suggest highlighting novelty of research here
Suggest checking English grammar; there were missing commas and wrong tenses
Wrong use of articles
1. Introduction:
Please rewrite sentence in line 31-32
Line 58-60: How is the biological activity of triazole related to or useful in PIM?
Line 66-67: Please check grammar. Which was used in this study, nickel(II) or cadmium(II)?
Line 71-72 can be merged with previous paragraph
2. Experimental:
Line 84-86: 1-alkyl-triazole derivatives (Table 1) were synthesized by Prof. A. Skrzypczak 84 (Poznan University of Technology, Poznan, Poland) by the alkylation reaction of 1,2,4-85 triazole.
Line 94-95: Please rewrite this sentence
Line 114-115: Do you mean that the feed contained mixture of Pb(II), Zn(II), and Cd(II)? Suggest stating it clearly
Line 118: Unit of surface area is cm2
Line 122-124: Please check grammar and rewrite this sentence
Line 128: RF is recovery factor or recovery coefficient? Please standardize
Please check equation (3) and (4); flux will be represented by J or I?
Was concentration of metal ions in feed measured at different times?
3. Results and discussion
3.1. Membrane study
3.1.1. Microscopic study of PIMs
Line 154: Figure 1 shows the AFM images of PIMs with …
Line 155-157: How does AFM show that the carrier was distributed homogeneously on the entire surface?
Please check Figure 1 caption
Line 173-177: Please check grammar and rewrite this sentence
3.1.2. Thermal stability of PIM doped with 1-alkyltriazole
Why only analysed thermal stability of PIM with 1-hexyltriazole? Are there any supporting information/ hypotheses to explain why degradation of CTA PIM occur in two steps? Do you expect difference in thermal stability between the PIM, PIM with 1-hexyltriazole before transport and PIM with 1-hexyltriazole after transport? Why?
3.1.3. FT-IR analysis of the PIM doped with 1-hexyltriazole
Similarly, why only performed FT-IR analysis on PIM with 1-hexyltriazole? Please add more discussion
3.2. Separation of Zn(II), Cd(II), and Pb(II) using alkyl triazole
Line 223-230 is not appropriate to be included in this section
Line 235-237: Please check grammar and rewrite this sentence
Check caption of Table 5
Line 261-262: Please check grammar; … with the highest rate …
Is the transport mechanism of metal ions across the PIM determined experimentally? In this study, how does the transport mechanism affects the separation of Zn(II), Cd(II), and Pb(II)? Suggest merging discussion from Section 3.3 here, instead of separating section 3.3.
Figure 6 requires legend
3.5. Recovery of metal
There seems to be inconsistency in the result obtained for recovery of Pb. PIM with carrier 3 resulted in the highest flux for all 3 metals, but the result here showed that PIM with carrier 1 resulted in the highest RF for Pb. Is there any reason for this observation? Please check
Author Response

(The authors gave the same response as above.)

Round 2
Reviewer 1 Report
Polish English
Author Response
Thank you for your time and comments. The text has been checked. The English language has been improved.
Reviewer 2 Report
The manuscript can be accepted.
Author Response
The text has been checked. The English language has been improved.
Reviewer 3 Report
Overall:
- There are still some noticeable grammatical mistakes (use of punctuations, etc.), use of English proofreading service is highly recommended
- Mistakes and inconsistencies can still be found throughout the article; please check carefully
Abstract:
Line 24-26 should be mentioned earlier and suggest that a brief concluding remark could be added at the end of the paragraph instead.
1. Introduction:
Line 32 -34: The use of heavy metals for industrial production is the main reason why these metals are present in wastewater, sewage and waste
Line 67-68: Please check grammar. Which was used in this study, nickel(II) or cadmium(II)? (Not yet corrected to cadmium(II) )
Line 73-80 is not suitable to be placed here
2. Experimental:
Line 98: Physical properties of the 1-alkyl-triazoles are listed in Table 1.
Line 144: RF is recovery factor or recovery coefficient? Please standardize (not yet corrected; please check under “Results and discussion”)
3. Results and discussion
Line 167: Not yet corrected; should be “Figure 1 shows the AFM images of PIMs with 1-alkyltriazole …”
Line 187 -191: Please correct grammatical errors; do you mean: “The roughness values determined for PIMs with carriers 1 - 3 are slightly higher than the roughness of PIM with the commercial carrier, D2EHPA (4.7 nm) that was used by Salazar-Alvarez [35]. However, the roughness values for CTA-o-NPPE-triazole membranes are comparable to the roughness found for PIM with alkylimidazole (3.7-7.2 nm) [16], and PIM with an imidazole derivative of azothiacrown 190 ethers (3.3 – 5.3 nm) [36].”
3.1.1. Thermal stability of PIM with 1-alkyltriazole
Why only analysed thermal stability of PIM with 1-hexyltriazole? Are there any supporting information/ hypotheses to explain why degradation of CTA PIM occur in two steps? Do you expect difference in thermal stability between the PIM, PIM with 1-hexyltriazole before transport and PIM with 1-hexyltriazole after transport? Why?
Any respond?
3.1.2. FT-IR analysis of the PIM with 1-hexyltriazole
Similarly, why only performed FT-IR analysis on PIM with 1-hexyltriazole? Please add more discussion
Any respond?
3.2. Separation of Zn(II) from Zn(II)- Cd(II)-Pb(II) mixture
Line 249-251: Please check grammar and rewrite this sentence (not yet corrected)
Table 5: Should be Pb(II) instead of Pd(II) (not yet corrected)
Suggest supporting Figure 5 with chemical equations showing the extraction and stripping reactions
Please check order of subheadings: 3.4. Recovery of metal
There seems to be inconsistency in the result obtained for recovery of Pb. PIM with carrier 3 resulted in the highest flux for all 3 metals, but the result here showed that PIM with carrier 1 resulted in the highest RF for Pb. Is there any reason for this observation? Please check
Any response?
Author Response
The answer is attached
